# In Silico Analysis of Possible Interaction between Host Genomic Transcription Factors (TFs) and Zika Virus (ZikaSPH2015) Strain with Combinatorial Gene Regulation; Virus Versus Host—The Game Reloaded

**DOI:** 10.3390/pathogens10010069

**Published:** 2021-01-14

**Authors:** Massimiliano Chetta, Marina Tarsitano, Laura Vicari, Annalisa Saracino, Nenad Bukvic

**Affiliations:** 1U.O.C. Genetica Medica e di Laboratorio, Ospedale Antonio Cardarelli, 80131 Napoli, Italy; marina.tarsitano@aocardarelli.it (M.T.); laura.vicari@aocardarelli.it (L.V.); 2Clinica di Malattie Infettive, Dipartimento di Scienze Biomediche ed Oncologia Umana, Università degli Studi “Aldo Moro” di Bari, 70124 Bari, Italy; annalisa.saracino@uniba.it; 3Genetica Medica, Azienda Ospedaliero Universitaria Consorziale Policlinico di Bari, 70124 Bari, Italy; nenad.bukvic@policlinico.ba.it

**Keywords:** Zika virus (ZIKV), TFs binding motifs, transcription factors (TFs)

## Abstract

In silico analysis is a promising approach for understanding biological events in complex diseases. Herein we report on the innovative computational workflow allowed to highlight new direct interactions between human transcription factors (TFs) and an entire genome of virus ZikaSPH2015 strain in order to identify the occurrence of specific motifs on a genomic Zika Virus sequence that is able to bind and, therefore, sequester host’s TFs. The analysis pipeline was performed using different bioinformatics tools available online (free of charge). According to obtained results of this in silico analysis, it is possible to hypothesize that these TFs binding motifs might be able to explain the complex and heterogeneous phenotype presentation in Zika-virus-affected fetuses/newborns, as well as the less severe condition in adults. Moreover, the proposed in silico protocol identified thirty-three different TFs identical to the distribution of TFBSs (Transcription Factor Binding Sites) on ZikaSPH2015 strain, potentially able to influence genes and pathways with biological functions confirming that this approach could find potential answers on disease pathogenesis.

## 1. Introduction

The Zika virus (ZIKV) is a positive-sense, single-strand RNA arbovirus belonging to the Flaviviridae family, genus Flavivirus, which includes other well known viruses, such as yellow fever, the dengue virus and the West Nile virus [1]. In 1947, ZIKV was originally isolated from a sentinel monkey, commonly found in the Zika forest in Uganda, east Africa. Shortly thereafter, other epidemiological studies evidenced the broad geographical spread of ZIKV throughout India, Southeast Asia and sub-Saharan Africa [2].

The infection causes a disease known as “Zika virus disease“, which is asymptomatic or paucisintomatic in approximately 80% of cases, and when symptomatic, adult populations experience low-grade fever, often accompanied by malaise, headache, dizziness, anorexia, conjunctivitis with retro-orbital pain, arthritis, transient arthralgia and maculopapular rash, which appear between 3 and 12 days after infection [2,3]. In some cases, neurological and autoimmune complications may occur with manifestations similar to those seen in Guillain-Barré syndrome. However, because fetal complications resulting from ZIKV infection contracted in pregnancy (first trimester) have been demonstrated [3,4], scientific interest has been increasing. In particular, Brazil was the first country where important fetal development issues were described, including microcephaly babies, with a high number of case notifications (2010: n = 153, 2011: n = 139, 2012: n = 175, 2013: n = 167 and 2014: n = 147), although high variability amongst different regions has been recorded [5].

ZIKV is spread by numerous Aedes species including Aedes aegypti, Aedes albopictus and Aedes henselli. In North America and Europe, Aedes albopictus is spreading as a new vector of ZIKV (spread from Gabon), as previously reported. The genus Aedes mosquito is common in warm climates; in the northern hemisphere it is present in North America and in different areas of the Mediterranean. Before 2007, a few cases of infections were reported at different places in Asian and African countries. Changes in the last years, due to human activities and climate changes, likely induced adaptive evolution mutation in the RNA genome of ZIKV [6]. Furthermore, during 2014–2015, ZIKV infection spread in South and Central America, the Caribbean regions and northeastern parts of Brazil. In Italy, the spread of these viruses is monitored by specific programs, as indicated in the letter of the Ministry of Health “National Surveillance Plan and response to arboviruses transmitted by mosquitoes 2018” [7]. 

Epidemic spread and the finding of autochthonous cases of ZIKV infection in various countries caused an international public health emergency. The health organizations therefore focused great attention on the potential risk of ZIKV transmission by transfusion of labile blood components (i.e., whole blood, packed red cells, fresh-frozen plasma, cryoprecipitate and platelets) [8]. Additionally, the transmission could also occur via breast milk, saliva and sexual intercourse. In fact, genomic ZIKV RNA was found in semen after six months in asymptomatic subjects and, in some cases, for at least ten weeks, in prostate, testicles and vaginal secretions [9]. 

Considering the impact of the ZIKV infection and its clinical consequences, studies are warranted to understand reasons underlining differences in clinical presentation, particularly with regards to occurrence of fetal abnormalities. 

The ZIKV genome is a single-stranded of ~10.500 nt with an RNA 5′ cap structure that codes one polyprotein that contains three structural peptides (capsid protein (C), envelope (E) and membrane (prM/M) glycoproteins) and seven nonstructural proteins (NS1, NS2A, NS2B, NS3, NS4A, NS4B and NS5) [10]. The translation of ZIKV RNA begins in the cytosol immediately after release from the virion. The N-terminal part of the polyprotein contains a signal localization at the level of the endoplasmic reticulum (ER) and promotes the association of ribosomes for the viral RNA translation. This determined a particular distribution of the protein domains: PrM, E, NS1 and some extended traits of NS2A, NS4A and NS4B which are localized intraluminally, while C, NS3 and NS5 are located on the cytoplasmic side of the ER, with different transmembrane sequences pre-sent in NS2A, NS2B and NS4B [10].

Furthermore, recent discoveries revealed how viruses use nascent RNAs to benefit their own gene expression; by adopting a parasitic lifestyle, they exploit parts of the host’s transcription pathways [11]. The viruses co-evolved many pathways to transcribe their own genetic material with consequent alteration of host transcriptional machinery [11,12]. ZIKV and other flaviviruses critically depend on the cellular secretory pathways, which transfer proteins and membranes from the ER through the Golgi and to the plasma membrane, for virion assembly, maturation and release. For this reason, the recruitment of transcription factors (TFs) could take place in cytoplasm of host cells; consequently, normal processes and “trafficking” like translocation into the nucleus could be blocked or sensitively decreased.

Herein we report on the innovative computational workflow allowed to highlight proposed interaction between host TFs and viral RNA (the first complete genome of Zika virus, has been deposited in the GenBank under the accession number KU321639) [13], which, as we hypothesized, could potentially lead to an effect comparable to haploinsufficiency. The phenomenon of haploinsufficiency is defined as reduced expression levels of involved protein as a result of a mutation in one allele of the corresponding TF, leading to a phenotypic effect. Depletion of cytoplasmic TFs, following viral infection, has recently been proposed and applied on the SARS-CoV-2 strain, demonstrating how the depletion of these TFs has generated the perturbation of different metabolic pathways with biological functions, confirming that this approach, though it is an in silico study, could bring useful insights also for ZIKV infection [14]. 

Furthermore, it is well known that during infection the accumulation of viral proteins is accompanied by a progressive global reduction in the production of host proteins, a phenomenon that has been termed “host shutoff” [11]. The functional annotation of putative in silico target genes of these motifs (TFs) are investigated to explore and understand processes which are under the complex and heterogeneous clinical presentation in ZIKV affected fetuses/newborns, as well as less severe presentation of infection in adults.

## 2. Materials and Methods

The analysis pipeline was performed using different bioinformatics tools available online and consists of four main steps (Figure 1) [15].

The entire ZikaSPH2015 strain was analyzed using a MEME-ChIP that performs a comprehensive motif analysis (including motif discovery) on large sets of sequences identified by ChIP-seq or CLIP-seq experiments on Human DNA (http://meme-suite.org/tools/meme-chip) [16].

All motifs identified were used as query for Tomtom (http://meme-suite.org/doc/tomtom.html), another tool of MEME suite that compared the motifs against a database of known motifs (i.e., HOCOMOCO human v11 full). Hocomoco is a complete collection of transcription factor binding models for humans, using a large-scale ChIP-Seq analysis. Tomtom ranked the motifs in the database and produced an alignment for each significant match, searching one or more query motifs against one or more databases of human target motifs (and their reverse complements when applicable). The report for each query was a list of target motifs, ranked by p-value in the order that the queries appear in the input file. The E-value and the q-value for each match were also reported. The q-value is the minimal false discovery rate at which the observed similarity would be considered significant. Tomtom estimated q-values from all the match p-values using the Benjamini and Hochberg method. By default, significance was measured by q-value of the match [17].

For all motifs queries, a list of TFs that contained the common conserved domain was obtained.

The list of TFs was loaded in STITCH (http://stitch.embl.de/), another software tool, to identify the most related correlation within query TFs set using a guilt-by-association approach. The bioinformatics tool used a large database of functional interaction net-works from multiple organisms, and each related TF is traceable to the source network used to make the prediction [18].

## 3. Results

The results of our in silico analysis allowed the identification of specific motifs on the ZikaSPH2015 strain able to bind and, therefore, according to our hypothesis, to sequester thirty-three different TFs. The distribution of TFBSs (Transcription Factor Binding Sites) on ZikaSPH2015 strain is shown in the Figure 2.

Using STITCH analysis, twenty-three of those TFs were connected in a gene regulatory knowledge network, while ten seemed to be out of this network and had no interactions (Figure 3).

This in silico approach allowed us to predict recruitment of some TFs directly related to different clinical manifestation (some of them syndromic) characterized by growth retardation, dysmorphic features, intellectual disability and others.

In particular, the analysis revealed a possible reduction of BCL11A (BAF Chromatin Remodeling Complex Subunit; OMIM 606557) TFs reported in association with intellectual developmental disorder with persistence of fetal hemoglobin (HbF) including microcephaly, down-slanting palpebral fissures, strabismus and external ear abnormalities [19].

The decrease of POU1F1 (POU Domain, Class 1, Transcription Factor 1; OMIM 173110) is implicated in growth retardation combined with pituitary hormone deficiency [20], reported in different patients affected by ZIKV, same as possible subtraction of ZNF214 (Zinc Finger Protein 214; OMIM 605015) involved in Beckwith-Wiedemann syndrome-BWSCR2 (Beckwith-Wiedemann Syndrome Chromosome Region-2) characterized by mild intellectual disability, obesity and characteristic dysmorphic features [21].

Indications regarding the relationship of PRDM6 (PR domain-containing protein 6; OMIM 616982) and ZFP28 (zinc finger protein 28; OMIM 616798) were put in evidence by our approach. PRDM6 is associated with isolated nonsyndromic patent ductus arteriosus, while ZFP28 plays an essential role in controlling gene expression during cardiac and vascular pathogeneses [22,23]. This approach endorsed an important interest considering involvement of MAF (MAF bZIP transcription factor; OMIM 177075), which is associated with juvenile-onset pulverulent cataract as well as congenital cerulean cataract (Cataract Multiple Types). Beside MAF, also two other regulators of several noncrystalline human-cataract-associated genes MAFG (MAF BZIP Transcription Factor G) and MAFK (MAF BZIP Transcription Factor K) which were also evidenced [24]. These observations could explain some ocular and heart defects manifestations frequently observed in ZIKV-affected babies/patients [22]. Furthermore, MAF haploinsufficiency was associated with brachycephaly and Fine-Lubinsky syndrome characterized by psychomotor delay, brachycephaly with flat face, small nose, microstomia, cleft palate, cataract and hearing loss [25], just as with Ayme-Gripp syndrome characterized by congenital cataracts, sensorineural hearing loss, intellectual disability, seizures, brachycephaly, a distinctive flat facial appearance and reduced growth [26].

Moreover, our analysis highlighted DUX4 (Double Homeobox 4; OMIM 606009), which, when depleted, is related to muscle weakness or atrophy, one of the main characteristics of Guillaume-Barrett syndrome [27]. Indeed, KLF5 (Krüppel-like factor 5; OMIM602903) belongs to the family of Krüppel-like TFs. The deficiency of this TFs is put in relation to unusual embryonic development, cardiovascular remodeling, inflammatory stress responses and intestinal development [28].

Another group of TFs, highlighted by this approach, were those involved in immune processes as they relate to immune specific regulatory elements. In particular, ZNF680 (Zinc Finger Protein 680), IRF4 (Interferon Regulatory Factor 4; OMIM 601900), SPI1 (Spleen Focus Forming Virus Proviral Integration Oncogene; OMIM 165170) and MEIS1 (Meis Homeobox 1; OMIM 601739), which are involved in differentiation and activation of macrophages or B-cells [29,30]. In this specific group of TFs, IRF8 (Interferon Regulatory Factor-8; OMIM 601565), SPIB (SPIB Transcription Factor; OMIM 606802) and TCF7 (Transcription Factor 7; OMIM 189908) have also been noticed. IRF8 is essential for development of monocytes, Plasmacytoid dendritic cells (pDCs) and Type 1 Conventional Dendritic Cells (cDC1s) [31]. SPIB is important in production of large amounts of interferon which block viral replication [32], while TCF7, predominantly expressed in T-cells, plays a critical role in natural killer cell and innate lymphoid cell development [33]. Our workflow allowd us to to take over involvement of AR (Androgen Receptor; OMIM 313700), which, when decreased in AR knockout mice, is associated with reduction of neutrophils function [34].

NFE2L2 (Nuclear Factor Erythroid 2-Like 2; OMIM 600492), ZNF250 (Zinc Finger Protein 250) and ZNF549 (Zinc Finger Protein 549) were activated in response to infection (i.e., Herpes Simplex Virus) and inflammation. The sequestration of these TFs makes the cells more susceptible to the virus infection [35,36].

By this approach, the indications on possible recruitment of different TFs involved in embryonic development was also underlined. In detail, the following have been indicated: LYL1 (LYM-phoblastic Leuke-mia-Derived Sequence 1; OMIM 151440) [37], SNAI1 (Snail Family Transcriptional Repressor 1; OMIM 604238) [38], ZFP28 (Zinc Finger Protein 28; OMIM 616798) [23], FEZF1 (FEZ Family Zinc Finger Protein 1; OMIM 613301) [39], HOXB4 (Homeobox B4; OMIM 142965), HOXB8 (Homeobox B8; OMIM 142963) [40], TCF7L1 (Transcription Factor 7-Like 1; OMIM 604652) [41], as well as those involved in brain development such as SMCA1 (Structural Maintenance Of Chromosomes 1A; OMIM 300040) [42], TBX3 (T-Box 3; OMIM 601621) [43], ZNF323 (Zinc Finger Protein 323; OMIM 610794) [44]. 

Finally, considering recent emerging evidences regarding the correlation between ZIKV and diabetes mellitus, our approach explains how the reduction of some TFs can be associated to pancreatic agenesis and/or disturbing its normal function. In particular PDX1 (Pancreas/Duodenum Homeobox Protein 1; OMIM 600733) is associated with early-onset of insulin dependent diabetes mellitus [45]; NKX6-1 (NK6, Homolog of Drosophila 1O; OMIM 602563) is implicated in development of insulin producing beta cells in the islets of Langerhans [46]; and TCF7L2 (Transcription Factor 7-Like 2; OMIM 602228) is involved in blood glucose homeostasis [47]. 

However, the analysis performed did not revealed a direct correlation with microcephaly.

## 4. Discussion and Conclusions

Based on the performed in silico analysis, it was possible to identify the occurrence of specific motifs in the ZikaSPH2015 strain. These motifs could be able to bind specific host (human) TFs, and, consequently, recruitment/subtraction could be provoked. In this manner, as we hypothesized, a haploinsufficiency-like effect is occurring, with consequences on involved target genes which are predicted by our in silico approach. Even though many studies are concentrated on the molecular mechanism of ZIKV infection, the exact mechanisms that induce the complex symptomatology, including microcephaly and brain anomalies in fetuses, remain unclear. An intricate network of TFs has been suggested by our analysis (Figure 3). Interestingly, the functional annotation of putative target genes revealed evidence that TFs are specifically enriched in biological networks involved in embryonic development or other functional clusters. All of these observations lead to a new insight, which should make possible the explanation of the complex and heterogeneous phenotype expression in ZIKV affected fetus/newborns, as well as less severe clinical picture of infection in adults. Indeed, this in silico analysis showed evidence that different TFs can regulate multiple target genes, and it provides original predictive computational data, paving the way as a testable hypothesis to further studies.

The Zika virus remains ambiguous in its role in development of severe microcephaly (primary microcephaly is characterized by significant microcephaly, below-2 SD for age, usually present at birth and always present before age one year and the absence of other congenital anomalies). In fact, specific TFs directly related to microcephaly were not evidenced by our approach. Our data is suggesting, according to recent observations, that microcephaly could be a result of the combined insufficiency of several embryonic neurological developmental TFs, able to produce a synergic effect on target genes which provoke this condition [48]; or, other factors could be engaged in this phenotypic expression. Namely, ZIKV, at least some strains, seems to have a specific neurotropism, and that is why major anomalies observed in fetuses and neonates include microcephaly, ventriculomegaly, diffuse calcifications, cerebral atrophy, signs of abnormal gyration, cortical development that might lead to severe mental retardation and motor disabilities together with visual and hearing impairments [49]. However, microcephaly might not always be observed, considering that a normal head circumference was observed in 20% of ZIKV congenital syndromes as reported by França et al. [50]. Furthermore, screening for microcephaly at birth is therefore not sufficient to detect congenital syndromes [51], considering that head growth deceleration has been reported in infants born with a normal head circumference, leading to development of microcephaly after birth [52]. The first genome sequence of ZIKV, from an autochthonous transmission in Brazil, was completed in 2016, and the authors concluded that it could be used as a starting point to study genome changes and gave an explanation why microcephaly was detected in Brazil and not in other countries [10]. Furthermore, Teng et al. [53] observed that ZIKV-related proteins might overlap the microcephaly associated proteins at the P53 signaling pathway, where P53 is the hub of the genetic regulatory network, causing viral-infection-induced cell death, thereby leading to a disorder characterized by abnormal brain development. Recently Chesnut et al. [54] summarized the research on in vitro and in silico models to study mosquito-borne flavivirus neuropathogenesis, prevention and treatment. This review article reporting on an in silico genetic screening approach performed by Kumar et al. [55] was utilized to elucidate that ZIKV-elicited congenital microcephaly may act through dysregulation of the retinoic acid response element (RARE). In addition, Pylro et al. employed an in silico analysis to investigate the role of both viral and human microRNA in congenital ZIKV microcephaly [56]. Finally, the observations regarding epigenetic and epitranscriptomic gene regulation has only recently begun to emerge, and it is well established that gene expression is not determined simply by the sequence information, but rather is subject to multiple levels of control [57].

In conclusion, the suggested pipeline is easy to access, and this easily reproducible approach could be used not only to understand the mechanism of infection and therefore the characterization of TFs involved, but also to identify metabolic networks suitable to identify tissue specific biomarkers and possible pathway dysfunction, which are useful for development of a possible vaccine or therapeutic strategy [14]. 

The application of these integrated analysis pipelines, which can be developed in a single user-friendly interface, could help support infection investigations by providing rapid information on multiple datasets analysis (i.e., genotype and transcriptome data), especially in times of crisis such as those currently in progress with SARS-CoV-2 [14].

Certainly, some of the differences observed in the phenotypic responses may be caused by interindividual genetic variations, but the identification of these regulatory elements should increase diagnostic sensitivity and accuracy of cohort classification, and therefore guide treatment [58].

## Figures and Tables

**Figure 1 pathogens-10-00069-f001:**
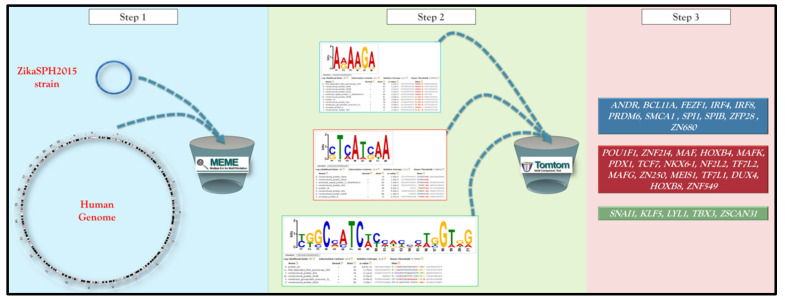
A schematic representation of the pipeline integrating the different analytical software. The details of the analytcal steps are described in the text (Materials and Methods section).

**Figure 2 pathogens-10-00069-f002:**
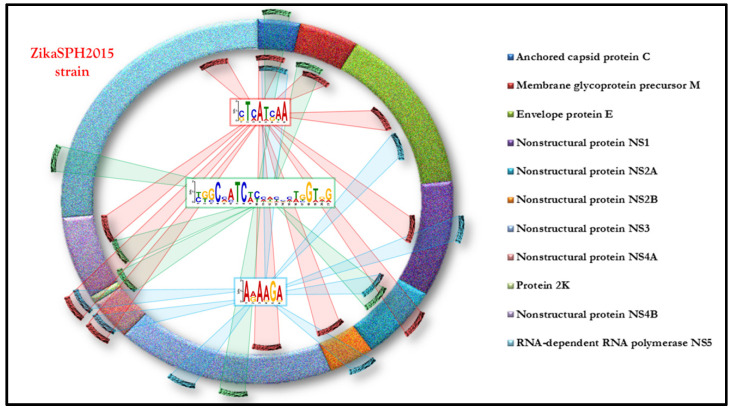
The distribution of TFBSs (transcription factor binding sites) on the ZikaSPH2015 strain.

**Figure 3 pathogens-10-00069-f003:**
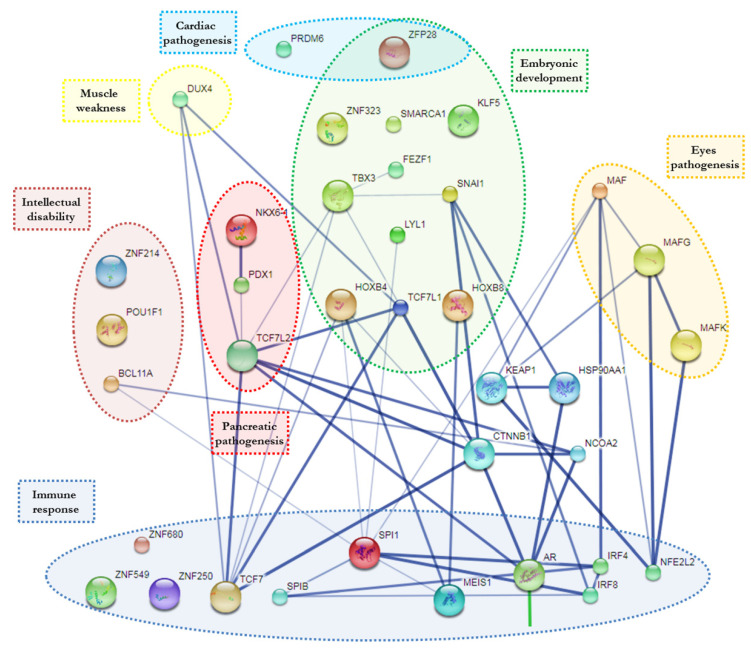
The scheme provides an overview of the TFs’ computational relationship analysis (based on STITCH tool) associated with each functional cluster.

## Data Availability

The data presented in this study are available on request from the corresponding author.

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
