# Peer review of "In Silico Analysis of Possible Interaction between Host Genomic Transcription Factors (TFs) and Zika Virus (ZikaSPH2015) Strain with Combinatorial Gene Regulation; Virus Versus Host—The Game Reloaded"

_pathogens, 2021, doi:10.3390/pathogens10010069_

Round 1

Reviewer 1 Report

Summary:

The manuscript titled “In silico analysis of possible interaction between host genomic transcription factors and Zika Virus (ZikaSPH2015) strain with combinatorial gene regulation; Virus versus host - the game reloaded” describes a new workflow for exploring how Zika virus (ZIKV) interacts with the host genome. More specifically the authors investigated how Zika virus (ZIKV) can sequester host Transcription Factors (TFs). This information in turn is valuable in understanding the pathogenesis of the virus in the host. Of most interest is in understanding the development of microcephaly clinical presentation in humans, which to date is still poorly understood. This manuscript is generally well written and concise. I have the following minor comments:

INTRODUCTION

Line 45: Should the phrase “In tropical region…” read in tropical regions?

Line 45-46: Italicize Aedes hensilli, aegypti and albopictus

Line 83: The word “subtraction” does not appear to follow the same font format as the rest of the manuscript.

MATERIALS AND METHODS

No comments

RESULTS

I suggest adding figure numbers to each figure so it is easy for the reader to identify the figures referenced in the text.

Line 129: “ZikaSPH2015” is italicized yet this name is not italicized earlier in the manuscript. Correct the formatting to be consistent throughout.

Line 200: “and…” is italicized. Please correct.

DISCUSSION AND CONCLUSIONS

Line 222: “ZikaSPH2015” is italicized yet this name is not italicized earlier in the manuscript. Correct the formatting to be consistent throughout.

Line 230: Please correct “These observation brings…”

Author Response

Reviewer 1

The manuscript titled “In silico analysis of possible interaction between host genomic transcription factors and Zika Virus (ZikaSPH2015) strain with combinatorial gene regulation; Virus versus host - the game reloaded” describes a new workflow for exploring how Zika virus (ZIKV) interacts with the host genome. More specifically the authors investigated how Zika virus (ZIKV) can sequester host Transcription Factors (TFs). This information in turn is valuable in understanding the pathogenesis of the virus in the host. Of most interest is in understanding the development of microcephaly clinical presentation in humans, which to date is still poorly understood. This manuscript is generally well written and concise. I have the following minor comments:

INTRODUCTION

Line 45: Should the phrase “In tropical region…” read in tropical regions?

The answer is “yes” and the sentence has been changed as suggested by the referee.

Line 45-46: Italicize Aedes hensilli, aegypti and albopictus  Aedes hensilli, aegypti and albopictus

The changes have been made as suggested by referee.

Line 83: The word “subtraction” does not appear to follow the same font format as the rest of the manuscript.

The changes have been made and the word “subtraction” has been cancelled (we used “recruitment”) as suggested by referee.

MATERIALS AND METHODS

No comments

RESULTS

I suggest adding figure numbers to each figure so it is easy for the reader to identify the figures referenced in the text.

The changes have been made as suggested by referee.

Line 129: “ZikaSPH2015” is italicized yet this name is not italicized earlier in the manuscript. Correct the formatting to be consistent throughout.

The changes have been made as suggested by referee.

Line 200: “and…” is italicized. Please correct.

The changes have been made as suggested by referee.

 DISCUSSION AND CONCLUSIONS

Line 222: “ZikaSPH2015” is italicized yet this name is not italicized earlier in the manuscript. Correct the formatting to be consistent throughout.

The changes have been made as suggested by referee.

Line 230: Please correct “These observation brings…”

The changes have been made as suggested by referee.

Reviewer 2 Report

This manuscript presents an in silico analysis that predicts Zika virus (ZIKV) protein - human transcription factor interactions. While this type of analysis is a helpful contribution to the Flavivirus research community, the manuscript could be improved by the inclusion of additional detail and findings.

Major Comments:
1.) Given the presentation of a new pipeline, some indication of the methodology used should be present in the abstract.

2.) The Materials and Methods section would benefit from substantial expansion about the principles behind the methods, and inclusion of details such as which human datasets were analyzed in the MEME Suite.

3.) The Results section would benefit from a table where the identified ZIKV TFBSs are presented. Which protein are they in? What is the amino acid sequence, or the amino acid positions? In other words, the regions depicted in Figure 2 need to be identified with more detail.

3.) The authors often discuss their findings in the context of congenital Zika syndrome or Zika encephalitis. Because these clinical presentations are only associated with Asian and Western Hemisphere strains (and NOT African clade strains), the analysis would be substantially improved by parallel analysis of an African clade strain as well as SPH2015. If the findings differ between the two, it would strongly bolster the relationship between findings and contribution to features of encephalitis and congenital Zika syndrome.

Minor Comments:
Line 35: the authors refer to "Zika fever" - the accepted clinical term for non-congenital impacts is "Zika virus disease".

Lines 42-43: Presenting the rates of microcephaly in 2015 and 2016 would provide important context to the rates from earlier years. Please add these.

Lines 45-49: This section can be tightened up by simply stating "ZIKV is spread by numerous Aedes species including Aedes aegypti, Aedes albopictus, and Aedes henselli." 

Lines 51-54: The timing should be clarified. While we now know that ZIKV was spreading in the Americas 2014-2015, the first case was diagnosed in 2015 and the Public Health Emergency of International Concern was not issued by the WHO until February 2016.

Lines 61-63: Add this to the previous paragraph, or delete it entirely.

Line 83: Define "TF" in the main text

Line 83-85: Please clarify that the goal of this work is to propose this suggestion as a potential process, and then indicate that this creates a testable hypothesis for future study.

Line 86: change "new direct interaction" to "proposed interaction"

Line 88-90: This passage is very unclear - please revise

Lines 91-94: Please indicate that the referenced study is also purely in silico

Line 120: change "motif's" to "motifs"

Line 242: Western Hemisphere/Asian lineage ZIKV strains have this tropism, but African lineage strains do not. Please indicate that the tropism is exhibited by some strains, but not necessarily all strains.

Line 266: "and this easy reproducible" should be "and this easily reproducible"

Author Response

Reviewer #2

Major Comments:
1.) Given the presentation of a new pipeline, some indication of the methodology used should be present in the abstract.

The changes have been made as suggested by referee.

2.) The Materials and Methods section would benefit from substantial expansion about the principles behind the methods, and inclusion of details such as which human datasets were analyzed in the MEME Suite.

The changes have been made as suggested by referee and the data set used for analysis has been introduced.

3.) The Results section would benefit from a table where the identified ZIKV TFBSs are presented. Which protein are they in? What is the amino acid sequence, or the amino acid positions? In other words, the regions depicted in Figure 2 need to be identified with more detail.

Explanation: The algorithm used by us, allowed to identify the nucleotide sequences consensus on the ZIKV complete genome, which is able to bind to the TFs as reported in the figure 1 (blue, red and green box). However, the algorithm did not provide information on the amino acid sequences or their positions.

3.) The authors often discuss their findings in the context of congenital Zika syndrome or Zika encephalitis. Because these clinical presentations are only associated with Asian and Western Hemisphere strains (and NOT African clade strains), the analysis would be substantially improved by parallel analysis of an African clade strain as well as SPH2015. If the findings differ between the two, it would strongly bolster the relationship between findings and contribution to features of encephalitis and congenital Zika syndrome.

Explanation: The analysis was conducted on the strain SPH2015 of which the complete sequence was deposited. This investigation was focused on the identification of possible TFs, which if depleted, could be traced to a phenotype of primary microcephaly.

The comparative analysis with other strains, certainly interesting, is currently not feasible because no other complete genomes have been deposited. This type of analysis and approach was performed on coronaviruses (pls. see reference 14).

Minor Comments:
Line 35: the authors refer to "Zika fever" - the accepted clinical term for non-congenital impacts is "Zika virus disease".

The suggestion has been accepted and changes has been made

Lines 42-43: Presenting the rates of microcephaly in 2015 and 2016 would provide important context to the rates from earlier years. Please add these.

We have not been able to address the concern raised by the reviewer. Considering that definition of congenital microcephaly, usually defined by measurement of occipital-frontal circumference, that is more than 2 SDs below the mean for age and sex or less than the 3rd percentile for age and sex. However the prevalence of microcephaly in the 15 states of Brazil with laboratory-confirmed Zika virus transmission was 2.8 cases per 10,000 live births, which was significantly higher than in the four Brazilian states without Zika virus transmission (prevalence 0.6 cases per 10,000 live births). Another review from North Eastern Brazil employing three different criteria showed markedly varying rates [Soares de Araújo J.S.R.C., Gomes R.G.S., Tavares T.R., Rocha dos Santos C., Assunção P.M. Microcephaly in northeast Brazil: a review of 16 208 births between 2012 and 2015. Bull World Health Organ. 2016]. In this review covering a period from 2012 to 2015, reported prevalence rates among 16,208 infants ranged from 2.1 to 8.0% based on the different criteria for congenital microcephaly used. It should be noted that in addition to microcephaly, Zika virus infection has been associated with other neurologic and brain abnormalities, which can be found in the absence of microcephaly [Leal M.C., Muniz L.F., Ferreira T.S., Santos C.M., Almeida L.C., Van Der Linden V. Hearing loss in infants with microcephaly and evidence of congenital Zika Virus Infection – Brazil, November 2015-May 2016. MMWR Morb Mortal Wkly Rep. 2016;65:917–919; Rasmussen S.A., Jamieson D.J., Honein M.A., Petersen L.R. Zika virus and birth defects - reviewing the evidence for causality. N Engl J Med. 2016; Ventura C.V., Maia M., Dias N., Ventura L.O., Belfort R., Jr. Zika: neurological and ocular findings in infant without microcephaly. Lancet. 2016;387:2502; de Fatima Vasco Aragao M., van der Linden V., Brainer-Lima A.M., Coeli R.R., Rocha M.A., Sobral da Silva P. Clinical features and neuroimaging (CT and MRI) findings in presumed Zika virus related congenital infection and microcephaly: retrospective case series study. BMJ. 2016;353;  de Oliveira-Szejnfeld P. Soares, Levine D., Melo A.S., Amorim M.M., Batista A.G., Chimelli L. Congenital brain abnormalities and Zika virus: what the radiologist can expect to see prenatally and postnatally. Radiology. 2016; 281:203–218; Russell K., Oliver S.E., Lewis L., Barfield W.D., Cragan J., Meaney-Delman D. Update: interim guidance for the evaluation and management of infants with possible congenital Zika Virus infection – United States, August 2016. MMWR Morb Mortal Wkly Rep. 2016;65:870–878].

We would like to addressed the concern raised by the reviewer and changes will be made if approve by the referee. Otherwise we are looking for a suggestion or advice.

Lines 45-49: This section can be tightened up by simply stating "ZIKV is spread by numerous Aedes species including Aedes aegypti, Aedes albopictus, and Aedes henselli." 

The suggestion has been accepted and changes has been made

Lines 51-54: The timing should be clarified. While we now know that ZIKV was spreading in the Americas 2014-2015, the first case was diagnosed in 2015 and the Public Health Emergency of International Concern was not issued by the WHO until February 2016.

Following the process described in the Brighton Collaboration Website

http://www.brightoncollaboration.org/internet/en/index/process.html, the Brighton Collaboration Congenital Microcephaly Working Group was formed in 2016. The composition of the working and reference group as well as results of the web-based survey completed by the reference group with subsequent discussions in the working group can be viewed at: http://www.brightoncollaboration.org/internet/en/index/working_groups.html.

Lines 61-63: Add this to the previous paragraph, or delete it entirely.

The suggestion has been accepted and this part was added to previous paragraph.

Line 83: Define "TF" in the main text

The suggestion has been accepted and changes has been made

Line 83-85: Please clarify that the goal of this work is to propose this suggestion as a potential process, and then indicate that this creates a testable hypothesis for future study.

We are grateful to the Referee for this comment and apologize for the poor clarity of this part. The Change has been made: “Indeed, this in silico analysis showed evidence that different TFs can regulate multiple target genes and provides original predictive computational data, paving the way as a testable hypothesis to further studies”.

Line 86: change "new direct interaction" to "proposed interaction"

The suggestion has been accepted and changes has been made

Line 88-90: This passage is very unclear - please revise

We thank the referee for this comment, the abstract section has been revised and rewritten to improve clarity as reported below:” The first complete genome of Zika virus, has been deposited in the GenBank under the accession number KU321639 [13]. Herein we report on the innovative computational workflow allowed to highlight, proposed interaction between host TFs and viral RNA, which, as we hypothesized, could potentially lead to an effect comparable to haploinsufficiency. The phenomenon of haploinsufficiency is defined as reduced expression levels of involved protein as a result of a mutation in one allele of the corresponding TF, leading to phenotypic effect. Depletion of cytoplasmic TFs, following viral infection, has recently been proposed and applied on SARS-CoV-2 strain, demonstrating how the depletion of these TFs has generated the perturbation of different metabolic pathways with biological functions, confirming that this approach, even though in silico study, could bring useful insights regarding ZIKV infection [14]”.

Lines 91-94: Please indicate that the referenced study is also purely in silico

The suggestion has been accepted and changes has been made (pls. see above)

Line 120: change "motif's" to "motifs"

The changes have been made

Line 242: Western Hemisphere/Asian lineage ZIKV strains have this tropism, but African lineage strains do not. Please indicate that the tropism is exhibited by some strains, but not necessarily all strains.

The suggestion has been accepted and changes has been made: “Namely, ZIKV, at least some strains, seems to have a specific neurotropism and that is why major anomalies observed in fetuses and neonates”.

Line 266: "and this easy reproducible" should be "and this easily reproducible"

The changes have been made